# Review of Dissolved CO and H_2_ Measurement Methods for Syngas Fermentation

**DOI:** 10.3390/s21062165

**Published:** 2021-03-19

**Authors:** Jie Dang, Ning Wang, Hasan K. Atiyeh

**Affiliations:** Department of Biosystems and Agricultural Engineering, Oklahoma State University, Stillwater, OK 74078, USA; jie.dang@okstate.edu (J.D.); hasan.atiyeh@okstate.edu (H.K.A.)

**Keywords:** syngas fermentation, dissolved CO measurement, dissolved H_2_ measurement

## Abstract

Syngas fermentation is a promising technique to produce biofuels using syngas obtained through gasified biomass and other carbonaceous materials or collected from industrial CO-rich off-gases. The primary components of syngas, carbon monoxide (CO) and hydrogen (H_2_), are converted to alcohols and other chemicals through an anaerobic fermentation process by acetogenic bacteria. Dissolved CO and H_2_ concentrations in fermentation media are among the most important parameters for successful and stable operation. However, the difficulties in timely and precise dissolved CO and H_2_ measurements hinder the industrial-scale commercialization of this technique. The purpose of this article is to provide a comprehensive review of available dissolved CO and H_2_ measurement methods, focusing on their detection mechanisms, CO and H_2_ cross interference and operations in syngas fermentation process. This paper further discusses potential novel methods by providing a critical review of gas phase CO and H_2_ detection methods with regard to their capability to be modified for measuring dissolved CO and H_2_ in syngas fermentation conditions.

## 1. Introduction

Synthesis gas (Syngas) fermentation is a novel, hybrid technique to produce biofuels and other bioproducts via a gasification–fermentation process based on traditional thermochemical conversion and biochemical fermentation [1,2]. Syngas is a mixture of CO, CO_2_, H_2_, N_2_, and minor gases such as CH_4_, NH_3_, H_2_S, and HCl produced from gasified carbonaceous materials including biomass, coal, and municipal wastes, or obtained from industrial waste gas streams [2,3]. The major components in syngas, CO and H_2_, are converted into ethanol, butanol, and other bioproducts through the acetyl-CoA pathway with extremely anaerobic microorganisms: *Clostridium ljungdahlii*, *C. autoethanogenum*, and *C. carboxidivorans* [2]. The overall biochemical reactions to produce ethanol, butanol, or acetic acids with both CO and H_2_ are shown as follows [3]:(1)$$3\mathrm{CO}+3H_{2}\to C_{2}H_{5}\mathrm{OH}+{\mathrm{CO}}_{2}$$
(2)$$7\mathrm{CO}+5H_{2}\to C_{4}H_{9}\mathrm{OH}+3{\mathrm{CO}}_{2}$$
(3)$$2\mathrm{CO}+2H_{2}\to {\mathrm{CH}}_{3}\mathrm{COOH}$$

This hybrid conversion process exploits the advantages of simplicity of gasification and product specificity of fermentation, while avoids many disadvantages in biochemical and thermochemical conversion techniques, such as incomplete utilization of biomass, complicated lignocellulose pretreatment, increasing concerns on competition with food supply, costly biological and chemical catalysts, harsh conditions during thermochemical conversion, and low product specificity [4,5]. However, many obstacles impede the industrial-scale commercialization of syngas fermentation, such as low productivity, unstable product specificity, and mass transfer limitations of CO and H_2_ [1,6].

One approach to overcome the above obstacles is to accurately control the dissolved CO (abbreviated as DCO) and dissolved H_2_ (abbreviated as DH) concentrations to improve overall process efficiency and operation stability [6]. CO and H_2_ are transformed to organic products through the acetyl-CoA pathway with reactions in sequence inside the cells of microorganisms (Figure 1).

The direction and extent of these reactions are determined by thermodynamics, which are affected by DCO and DH concentrations inside the fermenter [1]. Meanwhile, CO concentration has a negative effect on the hydrogenase enzyme activity and H_2_ utilization rate, which in turn influence the total carbon conversion efficiency of the fermentation process [2,6]. Therefore, appropriate DCO and DH concentrations are essential to make ethanol production thermodynamically favorable, and achieve high yields and product stability of syngas fermentation.

Accurate, inline, and fast-response sensors are the fundamental prerequisites for precise control of DCO and DH concentrations. To the best of our knowledge, specific DH sensors are currently available on the market for applications in power industry, e.g., membrane coated electrochemical sensors [7,8], while dedicated DCO sensors are not available so far. Meanwhile, limited research and applications were reported to measure DCO and DH concentrations. Fermenter mass transfer models were used to calculate the DCO and DH concentrations from headspace partial pressures of CO and H_2_ by gas chromatography (GC) and pressure transducer [1,9]. CO-myoglobin assays were reported as an offline method to determine DCO concentrations for research on the mass transfer process during syngas fermentation [10,11]. However, these methods are unsuitable for real-time, automatic DCO measurements due to their slow response and laborious manual operations. Besides applications in syngas fermentation, DCO measurement were reported in the determination of carboxyhemoglobin (COHb) as a CO exposure indicator [12] and the evaluation of CO’s signaling function in living tissues [13]. Practices of DH measurement were reported from the power industry [14,15], environmental science [16,17], and biochemical applications [18,19].

Besides the requirements for sensor accuracy, response, and inline measurements, the process of syngas fermentation has several additional requirements on the DCO and DH sensors. The most critical one is that the sensors should have excellent cross-selectivity between CO and H_2_ so that DCO and DH concentrations can be measured simultaneously. Many CO and H_2_ sensors, such as electrochemical [20,21], conductivity [22], or thermoelectric [23] sensors, fail to qualify this requirement since they cannot differentiate CO and H_2_ with great accuracy. The complex nature of components in the fermentation media, such as organic acids, alcohols, minerals, trace metals, and vitamins, require the sensors to have strong selectivity to these chemicals [24]. The strictly anaerobic fermentation process with acetogenic bacteria prohibits direct application of any sensors that require O_2_ for CO or H_2_ detection [2].

In summary, precise measurement of DCO and DH concentrations in the fermentation media remains a major challenge and obstacle for the development of economically sustainable syngas fermentation systems. The objective of this review is to examine current DCO and DH measurement methods for syngas fermentation. Various dissolved gas extraction techniques were also reviewed, which is often needed during dissolved gases measurements. Due to the disadvantages of currently available methods and lack of commercial DCO sensors, we provided a comprehensive review of gas phase CO and H_2_ detection methods and discussed their potential applications in DCO and DH measurements.

## 2. Dissolved CO Measurement Methods

### 2.1. Current Status of Dissolved CO Measurement Methods in Syngas Fermentation

Measurement of dissolved CO (DCO) in syngas fermentation medium is a challenging task because of the low CO solubility, interference from other chemicals, and conditions of syngas fermentation process. CO is sparingly soluble in water with a Henry’s law constant about 121,561 kPaL/mol at 37 °C, which results in a saturated concentration approximate to 23.25 mg/L (ppm) for pure CO under one standard atmosphere pressure [1,25]. Due to other major components (H_2_, N_2_, and CO_2_) in syngas, actual DCO concentration in fermentation medium is lower [6]. The fermentation medium contains many chemicals from syngas, nutrients for microorganisms, and fermentation products that may interfere with the DCO measurement [6]. H_2_ is the most significant interfering component as many CO detection mechanisms, such as electrochemical [26] or conductivity sensors [27], also respond to H_2_. Other chemicals, such as CH_4_, NH_3_, H_2_S, and HCl from syngas and alcohols and organic acids, may also affect the accuracy of certain CO detection mechanism. Lastly, syngas fermentation operates at specific conditions, which limit applications of some CO detection mechanisms. The microorganisms exploited in the fermentation are strictly anaerobic, so methods that require O_2_ during detection may be problematic [3]. Temperature and pH are also important for optimal cell growth of the microorganisms, which further limits the selection of CO detection mechanisms [1].

Currently, limited methods were introduced for DCO measurement in syngas fermentation. The CO-myoglobin assay method was reported to analyze gas–liquid mass transfer coefficients in syngas fermentation [28,29]. In our research, offline gas chromatography and fermenter gas mass transfer models were used to estimate the DCO concentration [30].

The CO-myoglobin assay method exploits metalloproteins, proteins with an iron ion cofactor such as hemoglobin and myoglobin, to detect dissolved phase CO by observing the changes in optical absorption spectra between CO-free and CO-bound metalloproteins [12]. DCO concentrations are obtained through predetermined fitting models between known DCO concentrations and optical absorption spectra of CO-bound metalloproteins [10,31]. However, the CO-myoglobin assay method requires complicated operation procedures and has a slow response (more than 30 min) [10]. In addition, the metalloproteins have limited lifespan, which implies that this method may not be appropriate for repeated, long-term dissolved CO measurements.

DCO concentration (mol/L) $C_{CO,L}$ in the bulk of liquid medium can be estimated through the liquid film mass transfer model with the help of gas chromatography [30]:(4)$$-\frac{1}{V_{L}}\frac{dn_{CO}}{dt}=\frac{k_{L,CO}a}{V_{L}}\left(C_{CO}^{*}-C_{CO,L}\right),$$
where $C_{CO}^{*}$ is the DCO concentration (mol/L) in the interface surface in equilibrium, which can be calculated from Henry’s law based on the headspace partial pressure of CO. $V_{L}$ is the volume of fermentation medium and a is the area (m^2^) of the gas–liquid interface surface. $k_{L,CO}$ is the liquid film mass transfer coefficient (L/m^2·^h), which is estimated beforehand. The molar rate of transfer (mol/h) $-\frac{dn_{CO}}{dt}$ represents the consumption of CO during the fermentation, which can be obtained by measuring CO partial pressure in the inlet and outlet gas flow with gas chromatography. DCO concentration can be obtained by solving the Equation (4) with the partial pressure data from gas chromatography. However, the accuracy of this method is highly related to the correctness of the mass transfer model and the mass transfer coefficient $k_{L,CO}$. Meanwhile the response of this method is slow due to the process to measure CO partial pressure from inlet, outlet, and headspace of the fermenter with gas chromatography.

Other DCO measurement methods were reported for medical or health applications, such as fluorescent optical sensors for CO imaging in tissues [32] and indirect methods for blood CO concentration measurement [33,34].

Fluorescent optical sensors fabricated with fluorescent proteins [13,32] and organic CO probes based on palladium catalyzed Tsuji–Trost reaction [35] were reported as a novel solution for in vivo CO imaging in animal tissues. The photoluminescence response triggered by reactions between CO and these fluorescent probes provides robust resistance to interference from other chemicals. These sensors were also reported with strong fluorescent response to dissolved phase CO in water-based solutions [32,36,37], which suggests that fluorescent optical sensors can have potential applications in DCO measurements for syngas fermentation. Simplicity and fast-response are the most appealing properties of fluorescent optical sensors [13,38]. However, fluorescent optical sensors may not be applicable for automatic, repeated DCO measurement, because their sensing reactions are conditional reversible with the aid of special reagents [13,38] or completely irreversible [36,37].

Determination of CO in blood can be indirectly measured with gas chromatography using a chemical CO extraction reagent. Reagents like formic acid [34] and ferricyanide [33] were reported to break down CO-bound metalloproteins to release gas phase CO from a fixed amount of blood samples. This approach circumvents the challenge of measuring DCO, but it is only applicable for blood CO measurement when metalloproteins exist. However, it is possible to use physical CO extraction methods for indirect DCO measurement in the syngas fermentation process.

Physical gas extraction methods are designed to extract dissolved gas by decreasing the partial pressure of the gas, such as the gas stripping technique [39], static headspace equilibration method [40], and vacuum extraction system [41,42]. Applications of semipermeable membranes in these methods were reported to achieve automatic, high-volume, rapid dissolved gas extractions [43,44,45]. In our opinion, membrane-aided vacuum extraction systems are the most practical DCO measurements methods with several automatic, in-house systems reported [43,44,46]. Dedicated dissolved gases measurement instruments, i.e., membrane introduction mass spectroscopy, were reported for applications in environmental science [47,48,49]. The successful applications of these gas extraction methods for gases with the same sparingly solubility, such as O_2_ and N_2_ [50], and noble gases, such as Ar, He, Ne, and Kr [43], suggest their potential applications in the DCO measurement. However, the sampling time of these gas extraction systems is around several minutes to hours, due to the time to establish new equilibrium in the gas extraction process, which are negatively related to the solubility of the target gas [45].

Despite the long sampling time and the complicated gas extraction system design, the introduction of physical gas extraction systems enables common gas phase CO sensor to indirectly measure DCO concentration. Current direct DCO measurement methods still have many obstacles to overcome for automatic, repeated DCO measurement, and the indirect DCO measurement methods helps to fulfill the current, urgent need from syngas fermentation while direct DCO measurement methods are still under improvement. A detailed review of gas phase CO detection mechanisms is performed to analyze their potential application in DCO measurement during syngas fermentation process, both directly and indirectly.

### 2.2. Potential Dissolved CO Measurement Methods

Contrary to the scarcity of DCO measurement methods, gas phase CO can be measured with various CO sensing methods [51], which can be roughly categorized into following groups: optical, acoustic, electrochemical, conductivity, work function type, and thermoelectric sensors. The review of current DCO measurement methods implies that an appropriate method to measure DCO in syngas fermentation should fulfill the following fundamental requirements:Highly sensitive to CO due to low CO solubility in the aqueous solution;Excellent selectivity to H_2_ and other chemicals in the fermentation medium;Sensor should not dramatically change the conditions of fermentation, such as anaerobic fermentation, pH, and temperature and;Sensor should be capable for automatic, rapid, and reversible measurement.

Therefore, the review of gas phase CO sensors will focus on their potential applications for direct or indirect DCO measurement according to their detection mechanisms while considering the abovementioned requirements for syngas fermentation process.

#### 2.2.1. Optical Sensors

Optical CO sensors utilize the spectral responses in ultraviolet, visible, and infrared range to detect CO. These CO sensors can be categorized into four groups: colorimetric sensors, fluorescent sensors, infrared sensors, and vacuum ultraviolet resonance fluorescence sensors.

##### Colorimetric Sensors

Colorimetric sensors exploit optical absorbance changes from interactions between CO and sensing materials for CO detection [52,53]. Metalloproteins [54], transition metal oxides [55], and transition metal (organic) complexes [56] are the mostly reported sensing materials in colorimetric CO sensors.

− Sensors Fabricated with Metalloproteins

Metalloproteins with iron cofactor, such as cytochrome c, myoglobin, and hemoglobin, were used as biological CO sensing materials by measuring their optical absorption spectra change from CO binding reactions [10,54]. The reaction between CO and cytochrome c was reported as irreversible, but myoglobin and hemoglobin demonstrated reversible color change between the brown to blood red color [54].

DCO measurements were reported using solutions of myoglobin and hemoglobin assay in glass cuvettes [10,12]. Immobilized metalloproteins in porous transparent sol–gel matrices were also reported as potential optical CO sensors [54]. However, myoglobin and hemoglobin were also sensitive to NO due to the formation of NO-myoglobin/hemoglobin compounds [54]. In addition, their interactions with other chemicals in syngas fermentation are unclear so far.

− Sensors Fabricated with Transition Metal Oxides

Thin films of transition metal oxides exhibit reversible absorption changes in visible–near infrared (VIS–NIR) spectra range in contact with CO under elevated temperatures [57,58]. The reported metal oxides for CO sensing include gold nanoparticles doped CuO [55], Co_3_O_4_, NiO, and Mn_3_O_4_ [59], SnO_2_ doped NiO and Co_3_O_4_ [60], CoO_x_ [61], and ZnO [58].

Several detection theories were proposed for the above reported metal oxides. For metal oxides with the p-type semiconducting property, such as Co_3_O_4_, NiO, and Mn_3_O_4_, the optical absorption spectra change is believed as the result of CO catalytic oxidation on these metal oxides [57,60]. The optical absorption spectra change of CuO sensing materials is believed from the plasmon absorption change of gold nanoparticles by CO absorption on the porous CuO (doped with gold nanoparticles) surface between 175 and 300 °C [57]. The reversible phase transition between CoO and Co_3_O_4_ in CO and CO-free air was reported as the detection mechanism of CoO_x_ sensing materials [61]. Transition metal oxides sensors, expect for Au-WO_3_ [62], have inferior selectivity to H_2_, because the selected metals also trigger the catalytic oxidation or absorption of H_2_ [57,59,60]. The selectivity of these sensors to other gases is also unclear.

Sensors fabricated with ZnO film doped on Au were reported with room temperature detection and fast response (in seconds) for CO in the range between 0.5 and 100 ppm; however, it relies on sophisticated spectrometer to measure the surface plasma resonance reflectance. Additionally, the sensing material demonstrates moderate response to H_2_, CO_2_, and NH_3_ [58].

− Sensors Fabricated with Transition Metal Complexes

Transition metal complexes, such as iron pincer complexes [63], rhodium complexes [38,64,65,66], and rhenium–iridium complexes [67], exhibit significant color changes sensible to naked eyes when they are exposed to CO. The color change is believed to be the result of the formation of CO coordinates by replacing the weakly bound donor ligands of the complexes with CO molecules [64,65,67]. Compared with the previous two types of optical CO sensing materials, transition metal complexes can detect CO under room temperature with measurement results sensible to human eyes [38,63,64]. The detection limit can be smaller than 1 ppm [68] with a typical response time ranged from seconds to hours depending on the selected metal complexes [38,64]. Fluorescent sensors fabricated with rhodium complexes demonstrated good selectivity to common gases, such as CO_2_, N_2_, and O_2_, but they were reported also as sensitive to NO and NO_2_ [38,65]. However, these research did not report selectivity of rhodium complexes to other gases or organic chemicals, such as H_2_ and alcohols [66,68,69,70]. The selectivity to other gases and alcohols for the other transition metal complexes were not reported so far.

− Potential Application of Colorimetric Sensors in Syngas Fermentation

Colorimetric sensors fabricated with metalloproteins, namely the CO-myoglobin assay, have been reported for direct DCO measurement in syngas fermentation; however, their detections require spectrometers to determine the absorption changes from CO-bounded proteins [10,28]. It is plausible for offline DCO measurement, but it will greatly reduce their potential value in automatic DCO measurement due to the complicated sample preparation procedures for spectrometers and the long equilibrium time (around 20 min [10]). Furthermore, spectrometers used to determine the absorption change greatly increase expense of these sensors.

Most transition metal oxides based colorimetric sensors work on an elevated temperature means that they can only paired with physical gas extraction devices to measure DCO concentration. Meanwhile, the absorption or reflectance change used for CO detection has to be measured with spectrometers [57,71]. Lastly, their sensitivity is determined by their sensing materials, which usually have weak selectivity to H_2_ and other gas components in syngas [57,60]. Therefore, these sensors can be considered unsuitable for syngas fermentation.

Colorimetric sensors using transition metal complexes have some pleasant potential in direct DCO measurement, such as the room-temperature naked-eye detection, good detection limit, and highly selectivity to common gases [68]. However, their applications are hindered by their hour-long recovery time [64,70] and unclear selectivity to H_2_ and other chemicals. The proposed sensing materials may take as long as 15 h [70] to reverse or be completely irreversible [67].

##### Fluorescent Sensors

Fluorescent sensors utilize the photoluminescence response triggered by reaction between CO and fluorescent probes for CO measurements in living tissues and aqueous solutions [32,72,73]. The transition metal complexes employed in some CO colorimetric, such as ruthenium, iridium, and rhenium, can also be used for fluorescent CO detection [56,68,74]. Besides transition metal complexes, fluorescent proteins [32], and organic fluorescent probes based on the palladium catalyzed Tsuji–Trost reaction [37,74,75,76] were also reported.

The fluorescent responses are the results of ligand replacement reactions with CO for fluorescent sensors using transition metal complexes [56,77]. Compared with corresponding colorimetric responses, the fluorescent responses with ultraviolet [36] or near-infrared [78] citation were much stronger. Fluorescent proteins utilize the molecular structure change from CO binding reaction of metalloproteins to activate the fluorescence dyes [32]. Organic fluorescent probes rely on the palladium catalyzed Tsuji–Trost reaction to replace the allyl group from the probes with CO molecules and to trigger the fluorescence activation in succession [74,76,79].

The detection limit of fluorescent sensors could reach the ppb level when using spectrometers for measurements [80,81]. Common gases, such as O_2_, NO_x_, CO_2_, H_2_S, and SO_2_, exhibited no response to fluorescent sensors based on the palladium catalyzed Tsuji–Trost reaction [36,74,81] and transition metal complexes [68]. However, no literature reported the selectivity of these sensors to H_2_ and alcohols, two common components in syngas fermentation media. The response time of fluorescent sensors was reported to range from twenty minutes to one hour [72,80,82], which is much slower than other CO detection methods. Although certain fluorescent probes can be reversed by using chemical agents [80,83], heating [84], or diazotization and iodization [79] for a repeated measurement; however, many fluorescent CO sensors utilize irreversible reactions for CO detection [69].

Despite the advantage of high CO sensitivity, robust selectivity, and low detection limit, current fluorescent CO sensors are not capable for rapid, repeated DCO measurement due to their slow response and poor reversibility. Meanwhile, the effect of ultraviolet radiation on the growth of microorganism remains unclear when fluorescent sensors are used directly in the fermenter. Lastly, the color (brown to black) of the fermentation medium may affect the correct recognition of fluorescent change. Currently, fluorescent sensors are more suitable for offline measurement of DCO concentration as an alternative of the CO-myoglobin assay method.

##### Vacuum Ultraviolet Resonance Fluorescence Sensors

Vacuum ultraviolet resonance fluorescence (VURF) sensors exploit the resonance fluorescence of $A^{1}\Pi \to X^{1}\Sigma$ transition located in the fourth positive bands of CO vacuum ultraviolet spectrum (around 150 nm wavelength) to detect CO [85,86]. CO concentration is calculated by comparing the measured ultraviolet fluorescence strength with the predetermined fluorescence strength of known concentrations during sensor calibrations [87].

Oxygen and water vapor are the major interference gases for VURF CO sensors, which both have strong absorption at the wavelength (around 150 nm) for CO fluorescence detection [86,87]. Due to ultraviolet photolysis of CO_2_ to CO, CO_2_ at high concentrations is an interference gas to VURF CO sensors; however, these sensors show no response to NO_2_ and NO [86]. Nevertheless, there is no report on possible interference from other common gases in the atmosphere.

VURF CO sensors were reported with comparable performance to tunable diode laser absorption spectroscopy (TDLAS) sensors in terms of response time, sensitivity, and detection limit [87]. The detection limit of VURF sensors was reported as 1-2 ppbv with a response time in several seconds [85].

Despite of these advantages in response time, sensitivity, and detection limit, the application of VURF in syngas fermentation is not applicable due to the component of CO_2_ in syngas and produced through fermentation.

##### Photoacoustic Sensors

Photoacoustic sensors, or optoacoustic sensors, exploit the acoustic waves generated by the absorption of incident radiation to measure CO concentrations [88]. The absorption of modulated incident radiation creates temporal temperature changes through the conversion of optical energy to heat [89]. Then, the temperature changes generate tiny pressure changes in the frequency of the modulated incident radiation, which can be detected with microphones or tuning forks as the photoacoustic signals [90,91]. The strength of the photoacoustic signal (mV), S, can be simplified as Equation (5) when the diversity of the pressure changes and the thermal/viscous losses are ignored [92]:(5)$$S=\frac{RW}{L}\left[1-exp\left(-\alpha CL\right)\right]$$
where W is the power (W) of the incident radiation, L is the cell length (cm), C is the sample concentration (mol·L^−1^), α is the gas optical absorption coefficient (cm^−1^∙mol^−1^∙L), and R is the cell responsivity (mV·cm·W^−1^).

Quartz enhanced photoacoustic sensors utilize quartz tuning fork as the acoustic transducer and optical energy collector [91]. The absorbed optical energy is accumulated in the quartz tuning forks instead of the sample gas in traditional photoacoustic sensors [93]. Therefore, quartz enhanced photoacoustic sensors have a much higher quality (Q) factor of the resonator and resonance frequency, which helps to eliminate background acoustic noise, remove design restrictions on gas cell, and improve detection limits [91,93].

Infrared sources in photoacoustic sensors include infrared emitters, LEDs, and lasers [94]. Lasers are the most popular option for photoacoustic sensors due to their high output power and narrow spectral bandwidth [91]. Gas discharge lasers [95], semiconductor diode lasers [92], and distributed feedback quantum cascade (DFB-QCL) lasers [96] were reported as the infrared sources.

Photoacoustic CO sensors were reported with a detection limit at the ppm level [95] while the quartz enhanced photoacoustic CO sensors could reach a ppb level using a short-path gas cell [91,96,97]. However, photoacoustic sensors face performance inconsistency from disturbances by environmental noises and short/long-term background signal fluctuations [94].

Photoacoustic sensors share many common features with infrared sensors except the approach to measure infrared absorption by acoustic waves instead of using photodetectors. Similar to infrared sensors, photoacoustic sensors can be used for indirect DCO measurement when they are connected with physical gas extraction devices.

##### Infrared Sensors

Infrared sensors exploits CO’s absorption at specific electromagnetic radiation ranges in the infrared region for CO detection [88,98]. Common infrared spectroscopy instruments include dispersive spectrometers, Fourier transform infrared (FTIR) spectroscopy, nondispersive infrared (NDIR) sensors, and tunable diode laser absorption spectroscopy (TDLAS) sensors [98]. NDIR and TDLAS sensors are dedicate instruments to measure the attenuation of infrared radiation at certain, narrow spectra range, which makes them the most prevailing CO infrared measurement techniques due to their simplicity, robustness, and accuracy [99,100,101]. Dispersive and FTIR spectrometers can measure multiple gases simultaneously as they are capable to obtain infrared absorption spectra in a wide spectral range [102,103,104].

Infrared sensors show the most promising potential for DCO measurement in syngas fermentation due to their detection principles, which measuring infrared absorbance/reflection at specific wavelengths for targeted compositions, e.g., DCO, while avoiding interferences from other chemicals. Hence, we provide a rather detailed review on the infrared sensors.

− Detection Mechanism of Infrared Sensors

Infrared absorption occurs when molecules demonstrate a change of dipole moment from molecular vibrations; thus, the wavelengths (frequencies) of the infrared absorption is related to the characteristics of the molecule structures [98]. Symmetric molecules, such as O_2_, H_2_, and N_2_, are infrared transparent because their molecular vibrations generate no electric dipole moment changes. CO molecules demonstrate “fingerprint-like” distinctive infrared absorption related to their asymmetric molecular structures [105].

Quantum theory states that infrared absorption occurs the energy of incident photons matches the difference between molecular vibrational energy levels [106]. The broad infrared absorption bands are composed of individual infrared absorption lines, whose wavelengths (frequencies) are determined by the molecular vibrations at discrete energy levels and other factors, such as overtone bands, Fermi resonance, coupling vibrations, and vibration-rotation [98]. CO molecules exhibit three infrared absorption bands around 4.6 µm, 2.3 µm, and 1.57 µm spectra range, which corresponds to its fundamental (at 4.6 µm), 1st overtone (at 2.3 µm), and 2nd overtone (at 1.57 µm) bands [107]. Since the overtone bands only absorb photons with higher energy, they exhibit less absorption strength several orders of magnitude smaller than that of the fundamental band [98]. Due to its relatively less interference from other chemicals, the fundamental infrared absorption band at 4.6 µm is the most promising wavelength for CO infrared detection [105,108].

The selectivity of infrared CO sensors is determined by the selected infrared absorption wavelength for measurements, especially for NDIR and TDLAS sensors. Interference occurs when other chemicals also absorb infrared radiation at the same wavelength for CO measurement. Therefore, NDIR sensors require narrowband infrared filters [100] or gas filter correlation techniques [109] to exclude infrared radiations outside the CO detection wavelength from broadband infrared sources.

The quantitative measurement of CO concentration (mol∙L^−1^), C, in the gas sample is calculated from the attenuation of transmitted infrared radiation strength according to the Beer–Lambert law [88]:(6)$$I=I_{o}e^{-\varepsilon CL},$$
where $I_{o}$ represents the intensity (W/sr) of incident infrared radiation, I represents the intensity (W/sr) of infrared radiation transmitted through the gas cell, ε represents the gas’s molar attenuation coefficient (cm^−1^mol^−1^∙L), and L represents the optical pathlength (cm) of the gas cell.

− Nondispersive Infrared Sensors

Nondispersive infrared (NDIR) sensors are designed to compare the incident and transmitted infrared radiation through gas samples to measure CO concentration [100]. Double beam design [110] and single beam design with the gas correlation filter [109] were the mostly reported sensor configurations (Figure 2). Double beam design utilizes a reference channel to determine the incident infrared radiation strength by an additional photodetector operated at a different, absorption-free wavelength [99]. Single beam design utilizes the gas correlation filter technique by introducing a mechanical chopper containing small gas cells with infrared transparent and target gas in the optical path. The rotation of chopper makes it possible to determine the incident infrared radiation strength with a single photodetector, which alleviates the performance shift problem in the double beam design [99]. Most NDIR sensors are built with single-pass, straight gas cell while U shaped or integrating sphere shaped gas cell were also reported [99,111].

Infrared radiation sources for NDIR sensors include Tungsten-halogen lamps [88], micromachined narrowband thermal emitters [112], and infrared light emitting diodes (LED) [113,114,115]. Tungsten lamp is heated to near 3000 K to emit broadband radiation with most of them resides in the visible and near-infrared region [116]. The micromachined emitters are excited to surface plasma states to emit narrowband infrared radiation at the designed wavelength range [112,117]. Infrared LEDs can emit narrowband infrared radiation in a spectral range between 1.7 and 4.8 µm under room temperature [115,118]. The output optical power of tungsten-halogen lamps and thermal emitters is much higher than that of infrared LEDs; however, their emission cannot be intrinsically modulated to higher frequency [99,119]. Infrared LEDs have a weak output optical power (in mW level) [120,121], but they can be intrinsically modulated to high frequency, thus simplifying the sensor structure [113]. Contrary to the high operation temperature of tungsten-halogen lamps and thermal emitters, infrared LEDs operate at room temperature, which requires no additional heatsinks [113].

Photodetectors for NDIR sensors include thermal detectors, such as thermocouples, thermopiles, or bolometers [99], and intrinsic quantum photodetectors, such as lead sulfide (PbS), lead selenide (PbSe), indium antimonide (InSb), and mercury cadmium telluride (MCT) photodetectors [122,123]. Extrinsic quantum photodetectors fabricated with Si and Ge were also used in NDIR sensors, but they need to operate at extremely low temperature [99,124]. MCT photodetectors are the most popular CO detectors due to their highest detectivity in 3–5 µm corresponding to CO’s strongest infrared absorption band at 4.6 µm [122,125]. They can be fabricated to detect infrared radiation in an operation temperature from 77 K (liquid nitrogen cooled) to room temperature [126].

− Tunable Diode Laser Absorption Spectroscopy (TDLAS)

TDLAS sensors measure the infrared absorption at a single infrared absorption line at very high resolution to provide high CO selectivity and sensitivity [101,127]. All three CO infrared absorption bands were utilized by TDLAS sensors: 4.6 µm [107,128], 2.3 µm [129,130], and 1.57 µm [131]. TDLAS sensors include direct absorption spectroscopy sensors [101], wavelength modulation spectroscopy sensors [127], and frequency modulation spectroscopy [101] (Figure 3). Direct absorption spectroscopy sensors are similar to NDIR sensors in structure by direct measuring the absorption from the sample to determine the CO concentration; thus they have to detect a tiny attenuation signal from sample in a strong background, which is shown as a dip corresponding to the gas absorption in the output signal as Figure 3a [127]. Wavelength modulation spectroscopy (WMS) TDLAS exploits a modulated laser control current, which modulated the emission wavelength of the laser source at around 100 Hz over the selected absorption line [101]. Gas concentration can be calculated using a lock-in amplifier on the second harmonic spectrum of the absorption line [88]. Thus, WMS TDLAS can improve the signal-to-noise ratio and provide a zero baseline signal [88]. In the case of frequency modulation spectroscopy (FMS) sensors, the frequency of laser control current is modulated to a very high frequency in the same magnitude as the line width of the target gas, which is usually higher than 100 MHz [132]. Therefore, FMS TDLAS has a lower 1/f noise and higher SNR for the lead-salt diode at the cost of high speed detector and lock-in amplifier [88,132], but they do not provide significant benefits for room temperature DFB lasers [88].

TDLAS sensors exclusively exploit lasers as their infrared source, including semiconductor lasers, doped insulator lasers, and quantum cascade (QC) lasers [101]. Using near-infrared lasers by the difference frequency generation (DFG) method can also generate lasers at the mid-infrared wavelength [133]. However, the output power from a DFG laser is inherently low and the highest emission is limited at up to 5 µm wavelength [134]. The distributed feedback QC lasers (QC-DFB laser) are the most popular infrared sources for trace gas measurements, because they have single-frequency emission spectrum with a high output power at room temperature [134,135]. The emission wavelength of QC-DFB lasers is determined by the size of quantum wells instead of the material band gaps; therefore, it is possible to design QC-DFB laser with specific emission wavelength by adjusting the size of quantum wells [136].

TDLAS sensors are usually designed with long optical pathlength due to lasers’ good directionality, which greatly improves their CO sensitivity [133]. The long pathlength can be achieved by using open-path optical design [128,130,137] or multiple-pass gas cells, such as the Herriott cell and White cell [138,139,140]. TDLAS CO sensors with optical fibers as the gas cell were reported in applications with very low CO sample volume [141,142].

− Infrared Spectrometers

Dispersive spectrometers and Fourier transform infrared (FTIR) spectrometers are common instruments to measure infrared absorption spectra. Dispersive spectrometers use diffraction gratings to disperse the incident infrared radiation to a wide spectral range and measure the absorption at individual wavelengths [143]. The dispersion of infrared radiation results in less radiation energy is used for the absorption measurement at a single wavelength, which increases the measurement time and reduces the single-to-noise ratio. Diffraction gratings also need frequent calibration to ensure their proper alignment.

Fourier transform infrared (FTIR) spectrometers are more advanced and accurate than dispersive spectrometers [144]. FTIR spectrometers utilize a Michelson interferometer (Figure 4) instead of the diffraction grating in dispersive spectrometers to generate the dispersed wavelengths. The moving mirror moves at fixed velocity repeatedly, so the beam from the moving mirror travelled at a different distance than the beam from the fix mirror. The recombined beam forms an interference pattern, called as interferogram, which can be used to obtain the single-beam spectrum by Fourier transform. Infrared absorption spectrum is obtained by comparing the single-beam spectrum from the background and the sample [98]. FTIR spectrometers are the most prevailing instruments in infrared absorption spectra analyses, because of their rapid measurement, good spectral resolution, and high signal-to-noise ratio [145].

− Potential Application of Infrared Sensors in Syngas Fermentation

Infrared sensors have a promising potential for DCO measurement in syngas fermentation due to their detection principles. The “fingerprint” infrared absorption based on the moment change from molecular vibrations provides an easy and estimable approach to determine the possible interference by examination of predetermined infrared absorption database. According to the absorption database [108], infrared CO sensors operated at 4.6 µm will not respond to other components in syngas, especially H_2_, and most chemicals in fermentation medium.

Meanwhile, the sensitivity of infrared sensors can be improved by increasing the optical pathlength while the detection limit is largely determined by the signal-to-noise ratio of detector and amplifier [88]. These characteristics distinguish infrared sensors to other CO sensors that rely on the chemical properties of their sensing materials to provide high sensitivity. Ambient operation temperature, rapid measurement speed, and no O_2_ requirement are other advantages of infrared sensors.

Due to the above advantages in CO detection, infrared sensors are particularly suitable for DCO measurement in syngas fermentation. However, due to the strong infrared absorption of water at 4.6 µm, 2.3 µm, and 1.57 µm [146], infrared sensors can only be paired with physical gas extraction devices to measure DCO concentration. The design of physical gas extraction devices determines the response and affects other performance, such as sensitivity and detection limit, of the infrared DCO measurement system. Among the above three types of infrared instruments, TDLAS sensors are the best option to measure extracted CO from the liquid sample. NDIR sensors, with inferior performance to TDLAS, can also be useful when the gas extraction devices are capable to extract CO from a large volume of sample in short duration [147,148].

#### 2.2.2. Acoustic Wave Sensors

Acoustic wave sensors exploit the frequency change of acoustic waves (mechanical vibrations) from interactions between CO and CO sensing piezoelectric materials to detect CO [149]. Based on propagation characteristics of the acoustic waves through the piezoelectric materials, acoustic wave sensors are categorized as bulk acoustic wave (BAW) sensors and surface acoustic wave (SAW) sensors [149].

Quartz crystal microbalances (QCMs) are the most commonly used BAW sensors [149]. A CO sensing layer is coated on piezoelectric materials to generate tiny mass change through CO oxidation [150,151] or CO absorption [152], which alters the vibrational frequency of the piezoelectric materials in succession. The change of vibrational frequency (Hz), dF, can be calculated as when the constants are determined for quartz [153]:(7)$$F=-2.3\times 10^{6}F^{2}\left(\frac{dM_{s}}{A}\right)$$
where $M_{s}$ is the mass of the coating materials (g), F is the oscillation frequency of the quartz (MHz), and A is the coating area (cm^2^). When the piezoelectric material has an oscillation frequency at the MHz level, QCM sensors are highly sensitive in CO measurements with high mass sensitivity [151].

The reported CO sensing coatings in QCM sensors include noble metals (platinum, palladium, and platinum-iridium alloy) [154], volatile metal oxides (HgO and Ag_2_O) [150,151], and various metal–organic complexes, such as the zinc crptand22 ligand complex [152], palladium acetamide complex [155], and nickel phthalocyanine complex [156]. Sensors using noble metals coatings exploit the heat generated from CO oxidation to increase the resonance frequency of the piezoelectric materials [154]. QCM sensors fabricated with HgO or Ag_2_O coatings utilize their reactions with CO at an elevated temperature to deposit a layer of mercury or silver metal on the surface of the microbalances [150,151]. The metal–organic complex coatings can directly absorb CO from ambient air and change the mass of the microbalances [152,155,156].

QCM sensors using metal–organic complex coatings are highly sensitive and were reported with a detection limit at the ppm level [152,155]. The response time ranged in seconds [156] to minutes [155]. However, QCM sensors fabricated with noble metals or volatile metal oxides also respond to other combustible or reducing gases [150,154]. QCM CO sensors fabricated with the zinc crptand22 ligand complex and nickel phthalocyanine complex were reported as sensitive to NO_2_ [152] and SO_2_/NO_2_ [156], respectively. QCM sensor with a palladium acetamide complex coating was reported free of interference from H_2_, SO_2_, and H_2_S; however, the CO absorption on the coating is irreversible [155].

Surface acoustic wave (SAW) sensors have the propagation of mechanical waves confined to the surface of the medium. Therefore, SAW sensors require patterned thin-film interdigital transducers coated on the surface of piezoelectric materials to detect the acoustic waves [149]. The operation frequency of SAW sensors is much higher (up to 1 GHz) than that of BAW sensors, which results in a better sensitivity [149].

The CO sensing layer can be coated on the top of thin-film interdigital transducers or between the transducers at the same horizontal level [149]. The reported CO sensing coatings include ZnO [157], WO_3_ [158], graphene nano-sheet [159,160], cobalt corroles [161], and polyaniline (PANI) [162]. However, most reports did not provide results on sensor selectivity to reducing gases, such as H_2_.

In summary, the detection mechanism of acoustic wave sensors demonstrates that they cannot be used to direct measure DCO concentration in aqueous solution. The application of acoustic wave sensors is hindered by the selectivity issue and irreversible detection reaction from current CO sensing coatings, which makes them less attractable than infrared or photoacoustic sensors for indirect DCO measurement.

#### 2.2.3. Electrochemical Sensors

Electrochemical sensors measure changes in electrical properties from CO’s electrochemical reactions occurred at the sensors’ electrodes. These are classified as amperometric sensors and potentiometric sensors based on the electrical properties employed in detection [163,164].

##### Amperometric Sensors

Amperometric sensors measure the electrolysis current between electrodes to determine CO concentrations [164]. A constant [26] or variable [165] voltage is applied between the electrodes to facilitate the electrochemical reactions on the surface of electrodes. According to the Faraday’s Law, the measured current, which reflects the electrochemical reaction rate, is proportional to the CO concentration when the sensors are operated under appropriate diffusion-limited conditions [26,164].

Amperometric sensors consisted of three basic components: electrodes, an electrolyte, and a gas permeable layer. Most sensors have a three-electrode configuration with one working electrode, one counter electrode, and one reference electrode. Electrolyte, in the aqueous solution or solid state, functions as an ion transportation medium between electrodes. Early amperometric sensors were reported using the aqueous electrolyte such as sulfuric acid [26]. Solid state electrolyte materials were introduced to build miniaturized sensors, such as the Nafion^®^ membrane [166,167], zirconium phosphate film [168], NASICON (Na_3_Zr_2_Si_2_PO_12_) [169], and yttria-stabilized zirconia [170]. A gas permeable layer, usually made with Teflon^®^ materials, is used to cover the electrodes and the electrolyte to control the gas diffusion and to prevent leakage of the electrolyte [164].

The electrochemical reactions in amperometric CO sensors involve the oxidation of CO at the working electrode and the reduction of O_2_ at the counter electrode [171]:(8)$$\mathrm{CO}+H_{2}O\to {\mathrm{CO}}_{2}+2H^{+}+2e^{-}$$
(9)$$\frac{1}{2}O_{2}+2H^{+}+2e^{-}\to H_{2}O$$

The overall reaction is:(10)$$\mathrm{CO}+\frac{1}{2}O_{2}\to {\mathrm{CO}}_{2}$$

The electrodes materials of amperometric CO sensors determine sensor CO selectivity. Sensors fabricated with platinum electrodes also sensitive to other reducing gases, such as NO_x_, H_2_, and hydrocarbons, due to platinum also can catalyze oxidation reactions of these gases [21,164]. Novel electrode materials, such as ruthenium-platinum alloy [172], gold-nanoparticles doped platinum [173], and multiwall carbon nanotubes grafted polydiphenylamine [165], were reported for better CO selectivity to these reducing gases.

Advantages of amperometric CO sensors include high sensitivity, low limit of detection (0.01 ppm [165]), rapid measurement, and linear response [164]. However, the performance of solid state electrolyte sensors are influenced by relative humidity in the gas flow due to the requirement of water in detection [21,168].

##### Potentiometric Sensors

Potentiometric CO sensors measure the electromotive force (emf) or potential difference from CO oxidation at the electrodes to determine CO concentration [174,175]. The structure of potentiometric sensors is similar to a galvanic cell with two electrodes immersed in electrolyte as the anode and cathode [175].

For CO detection, the measured emf is generated from a mixed potential established through the reduction of O_2_ and the oxidation of CO [175]:(11)$$\frac{1}{2}O_{2}+2e^{-}\to O^{2-}$$
(12)$$\mathrm{CO}+O^{2-}\to {\mathrm{CO}}_{2}+2e^{-}$$

The overall reaction is:(13)$$\mathrm{CO}+\frac{1}{2}O_{2}\to {\mathrm{CO}}_{2}$$

Similar to amperometric CO sensors, the selectivity of potentiometric CO sensors is determined by their electrode materials. Thus, CO sensors fabricated with platinum or gold electrodes have inferior selectivity to H_2_ and hydrocarbons [20,174].

Sensors with better CO selectivity were reported by adding specific CO sensitive catalysts to the platinum electrodes, such as CuO or ZnO catalysts [176] or using various metal oxides electrodes, such as perovskite-type oxides (LaMO_3_) [20], mixture of CdO and SnO_2_ (CdO-SnO_2_) [177], and Co_3_O_4_ with gold nanoparticles (Au-Co_3_O_4_) [178]. The best CO selectivity and sensitivity were reported at an operation temperature around 400–700 °C for above materials. However, their CO sensitivity and selectivity to H_2_ were affected by relative humidity of gas flow [169,179].

Electrochemical sensors have been used in the measurement of dissolve phase H_2_ and O_2_ by coating the electrodes and electrolyte with a selective permeable membrane. Nevertheless, the electrochemical detection of CO requires O_2_ as a reagent, which is impracticable in the anaerobic syngas fermentation process due to the challenge to introduce the O_2_, whether directly to the fermenter or indirectly to the sealed extracted gas samples. Meanwhile, electrochemical sensors do not demonstrate better sensitivity, selectivity, and response time than infrared CO sensors when used as the CO sensing unit for indirect DCO measurement system. Therefore, electrochemical sensors are impractical for both direct and indirect DCO measurement in syngas fermentation.

#### 2.2.4. Conductivity Sensors

Conductivity CO sensors utilize the reversible conductance changes from interactions between CO and specific semiconductors, namely metal-oxide semiconductors and conducting polymer semiconductors, to detect CO [180,181].

##### Metal-Oxide Semiconductor Sensors

Common metal-oxide semiconductors reported for CO detection include SnO_x_ [181], α-Fe_2_O_3_ [27], and In_2_O_3_ with Rb_2_O/Co_3_O_4_ catalysts [182,183]. The conductivity changes of most metal oxides can be explained as the result of the electrons trapping and the band bending of the oxides associated with reactions between CO and O_2_ [183,184]. Some metal oxides have different CO detection theories, such as CO absorption on sensing material [185] and collapsed charge order restoration with CO contacts [186].

Metal-oxide semiconductor CO sensors are mature, low cost CO measurement methods with very simple structure [187]. However, they have poor selectivity to other reducing gases, such as H_2_, NO_2_, and CH_4_ [27,188]. Doping the bulk semiconductors with specific metal oxides, such as Co_3_O_4_, CuO, ZnO, NiO, Y_2_O_3_ [188], MoO_3_ [189], or pure metals, such as platinum, lead, titanium, and copper [27,190] were reported with improved CO sensitivity and selectivity. Modification of the surface nanostructure of some metal-oxide semiconductors, such as SnO_2_, is another approach to improve CO sensitivity and selectivity [191].

Many metal-oxide semiconductors are usually heated to an elevated temperature (200–400 °C) for the best CO sensitivity [184,188]. Room temperature metal-oxide CO sensors usually have poor selectivity to gases like NO, H_2_, and CH_4_ [22]. Meanwhile, CO sensitivity of these sensors is highly influenced by relative humidity [184,192,193].

##### Conducting Polymer Semiconductor Sensors

Conducting polymers refer to several organic large molecules, such as polypyrrole (PPy), polyaniline (PANI), and polythiophene (PTh) [180,194]. PANI is the mostly reported conducting polymer for CO detection [195,196] and is often doped with catalysts to enhance CO sensitivity and selectivity, such as HCl [197], Co_3_O_4_ [198], and TiO_2_ [199].

The detection mechanism of PANI can be explained as the result of the electrons transfer between CO and the polymer [198]. PANI is a P-type semiconductor with majority charge carriers of holes [180]. CO molecules extract electrons from PANI, thus increase the number of charge carrier and the conductivity of PANI [197]. Other theories were also proposed, such as physical absorption of CO to PANI [200] and redox reaction between CO and PANI [196].

High sensitivity and rapid response at room temperature are the advantages of conducting polymer semiconductors over the metal-oxide semiconductors [180]. The response time for CO at low concentration was reported within several seconds [194,197]. Vacuum deposited PANI polymer was reported with a detection limit of 0.02 ppm [200]. The selectivity of conducting polymer CO sensors is determined by their material compositions and the structure of the polymers [194]. Conducting polymer semiconductor sensors were reported with good selectivity to H_2_ [197], liquefied petroleum gases, and CH_4_ [198]. However, certain conducting polymer CO sensors demonstrate dependence on relative humidity [198,201].

For syngas fermentation applications, most metal-oxide semiconductor sensors are not feasible due to their requirement of O_2_ and poor selectivity at room temperature. Conducting polymer semiconductor sensors based on electrons transfer or physical absorption are applicable for indirect DCO measurement after further experiments to settle issues of selectivity, repeatability, and influence from relative humidity.

#### 2.2.5. Work Function Type Sensors

Work function type CO sensors include field effect transistor (FET) sensors [202], metal-oxide-semiconductor (MOS) capacitor sensors [203], and the Schottky diode sensors [204].

##### Field Effect Transistor Sensors

FET sensors are fabricated by replacing the gate materials in a normal FET with CO sensing metals or metal-oxide semiconductors, whose carrier concentration can be altered in contact with CO [205]. The alternation of gate voltage threshold by carrier concentration change is used to determine CO concentration [202,206]. MOS, metal-insulator-semiconductor (MIS), and suspended gate type of FET CO sensors were reported using palladium [202] or Pd-PdO mixture [205], porous Pt-SnO_2_ mixture [206] or Pt-WO_3_ mixture [207], and Al_2_O_3_ [208] as the gate materials, respectively.

The sensitivity and selectivity of FET CO sensors are determined by the gate material and/or the gate surface nanostructure [207,209]. Current gate materials were reported with inferior selectivity to H_2_ [203], CH_4_ [209], and ethanol [205]. Their response time ranged from 75 s [207] to several hundred seconds [203]. FET CO sensors usually require an elevated temperature for better sensitivity. The optimal temperature depends on the materials, ranging from 75 [207] to 180 °C [202]. Room temperature FET sensors were reported using Al_2_O_3_ gate materials [208].

##### MOS Capacitor Sensors

MOS capacitor sensors exploit capacitance changes from CO absorption on palladium gate [210] or CO oxidation catalyzed by the Pt-FeO_X_ mixture gate [211] to detect CO. MOS capacitor sensors using palladium have inferior selectivity to H_2_ [209] while the selectivity of MOS capacitor sensors with Pt-FeO_X_ gate material to H_2_ were not reported [211].

##### The Schottky Diode Sensors

The Schottky diode is a special type of diode with its junction formed with metal and semiconductor. Therefore, it can detect CO when the junction materials are fabricated with either CO sensing metals or metal-oxide semiconductors. Reported junction materials include polyaniline (PANI) [204], ZnO [212], SnO_2_/TiO_2_ [213], ITO (Indium tin oxide) [214], AlGaN-GaN [215], Pt-GaN [216,217], and Au-GaAs [218].

CO concentration is measured through the change of current–voltage (I–V) characteristics of the Schottky diode, which is generated from the conductivity change of the junction materials [214]. The conductivity change is the result of either CO oxidation by the metal oxides such as ZnO, SnO_2_, and TiO_2_ [212,213] or CO absorption at the semiconductors [204,214] and the noble metals [216,218].

The selectivity of the Schottky diode CO sensors is largely determined by their junction materials. CO sensors with Au-GaAs junctions were reported also being sensitive to NO [218] while Pt-GaN junctions were reported to sensitive to H_2_ [216]. However, no test results were reported for sensors fabricated with other junction materials. The response time of the Schottky diode CO sensors was reported within a few seconds [214,219]. These sensors can operate at room temperature, but those fabricated with metal oxides have the best CO sensitivity at elevated temperature [212,213].

The detection mechanisms of work function type CO sensors determine that they are almost unworkable for syngas fermentation application. Sensors based on CO catalytic oxidation need to address the challenge to introduce O_2_ to a strictly anaerobic environment while sensors based on CO absorption mechanism demonstrate selectivity issue to H_2_.

#### 2.2.6. Thermoelectric Sensors

Thermoelectric sensors utilize the catalytic oxidation of CO to measure CO concentration [23]. The heat from CO oxidation creates a temperature gradient between the cold side and the hot side of the sensor’s junction, which generate a voltage signal for CO measurements according to the thermoelectric (Seebeck) effect [220]. A linear relationship exists between the voltage and the logarithmic scaled CO concentration [221,222].

The reported catalysts in thermoelectric CO sensors include a mixture of SnO_2_ and Co_3_O_4_ doped with gold [221], Co_3_O_4_ doped with gold [23,223], TiO_2_ doped with gold [222], CoO doped with gold, platinum, or palladium [220], Co_3_O_4_ and CeO_2_ mixture [224], and CeO_2_ and ZrO_2_ mixture [225]. Some of these catalysts were reported with a good selectivity to H_2_, CH_4_, and alcohols [23,220,221,224]. The detection limit of thermoelectric sensors could reach 1 ppm at 200 °C [220]. Room temperature thermoelectric CO sensors were reported with an inferior detection limit (5000 ppm) due to their mediocre CO catalyzing capability at lower temperature [223].

Thermoelectric CO sensors require an elevated operation temperature for high sensitivity [220,224], which limits their applications in syngas fermentation process with the presence of combustible gases. The signal strength of thermoelectric sensors is usually around several microvolts [220,221,222]. Hence, thermoelectric CO sensors are usually paired with highly sensitive data acquisition devices, which increase their total system cost.

Since thermoelectric CO sensors based solely on the catalytic oxidation to measure CO concentration, they are not suitable for the strictly anaerobic syngas fermentation process. The difficulties to introduce O_2_ outweigh the advantages of thermoelectric CO sensors.

### 2.3. Summary of the Potential Dissolved CO Measurement Method

Review of current dissolved CO measurement methods suggests that there are two plausible approach for the DCO measurement. The direct approach, which is to measure DCO concentration with specific CO detection mechanisms, such as the myoglobin assay and fluorescent probes. The indirect approach, which is to employ chemical or physical gas extraction devices to measure extracted gas phase CO. Thus, a comprehensive review of common CO sensors was performed (Table 1) to examine their potential applications in DCO measurement based on detection mechanisms and compatibility with the syngas fermentation process.

The review suggests that direct measurement of DCO concentration might be achieved using colorimetric sensors fabricated with transition metal complexes, fluorescent sensors with transition metal complexes, fluorescent proteins, or palladium catalyzed Tsuji–Trost reaction. However, the drawbacks of colorimetric sensors and fluorescent sensors in the reversibility issue, such as a long recovery time [70], reagents [73], and heating [84] aided reversibility, indicate that these sensors are most appropriate for offline, disposable DCO measurement.

Indirect measurement of DCO concentration circumvent the challenge to measure DCO in the fermentation medium by using a physical gas extraction system at the cost of a slow response and high system complexity. Infrared and photoacoustic sensors are the most appropriate methods for indirect DCO measurement mainly because of their detection mechanisms, which measure CO infrared absorption to determine its concentration [97,105]. Electrochemical, conductivity, work function type, and thermoelectric sensors exploits detection mechanisms based on the oxidation of CO, which makes them impractical in syngas fermentation due to the challenges to introduce O_2_ to the extracted gas samples. Other advantages of infrared and photoacoustic sensors, such as fast response, high sensitivity, low detection limit, and free of interference [88,105,108], also makes them the best option in the indirect DCO measurement.

Tunable diode laser spectroscopy (TDLAS) sensors and quartz enhanced photoacoustic (QEPAS) sensors are the best candidates due to their high sensitivity with long optical path design and detection limits at the ppb level [91,101]. Thus, the time required for gas extraction can be shorter when only a tiny amount of liquid sample is analyzed. Nondispersive infrared (NDIR) sensors and photoacoustic sensors were reported with inferior detection limits at the ppm level [100,226], but they can be applicable when the gas extraction system can handling large volume of sample in short time, such as system built with a hollow fiber membrane contactor [147,227].

**Table 1 sensors-21-02165-t001:** Comparison of CO detection mechanisms in syngas fermentation application.

Sensor Type	Direct DCO Measurement Potential	Sensitivity/Detection Limit	Selectivity	Measurement Conditions	Other Challenges
Colorimetric: transition metal oxide	No report	High, 0.5 ppm [58]	H_2_	Room and elevated temperature, May require O_2_	Need spectrometer for signal analysis
Colorimetric: metallic proteins	CO-myoglobin assay	Moderate, unclear detection limit	NO	Room temperature	Proteins have limited lifespan, Slow recovery time
Colorimetric: chromogenic probes	Probes in solutions was used in CO detection	High, 11 ppm [38]	NO_X_, Require additional selectivity test	Room temperature	Color may be shaded by fermentation media
Fluorescent	Living tissue and in aqueous solution	High, ppb level [81]	Unclear for H_2_, Require additional selectivity test	Room temperature	Irreversible response, Need UV excitation of fluorescent probes
Nondispersive infrared	No report	High, ppm level [100]	HCN at 4.6 µm	Room temperature	Need infrared database analysis for proper detection wavelength
Tunable diode laser absorption spectroscopy	No report	Very High, ppb level [133]	HCN at 4.6 µm	Room temperature	Require mid-infrared laser for max sensitivity
Spectrometer and FTIR	No report	High, unreported detection limit	HCN at 4.6 µm	Room temperature	Complicated and expensive
Vacuum ultraviolet resonance fluorescence	No report	High, 1 ppb [85]	CO_2_, water vapor	Room temperature	Complicated system, Measurement range is small
Photoacoustic	No report	Very high, ppb level [97]	HCN at 4.6 µm	Room temperature	Require mid-infrared laser for max sensitivity
Bulk acoustic wave	No report	High, 0.91 ppm [152]	Depend on sensing materials	Room and elevated temperature, May require O_2_	Irreversible detection Limited CO exposure time
Surface acoustic wave	No report	Moderate, 25 ppm [158]	Unclear	Room and elevated temperature. May require O_2_	
Amperometric	No report	High, 0.01 ppm [165]	Reducing gases NO_x_, H_2_, and hydrocarbons	Room and elevated temperature. Require O_2_	Measurement affected by relative humidity
Potentiometric	No report	High, 1 ppm [169]	Reducing gases NO_x_, H_2_, and hydrocarbons	Best sensitivity at elevated temperature, Require O_2_	
Conductivity: Metal oxide semiconductor	No report	Moderate, 6-18 ppm [228]	Reducing gases NO_x_, H_2_, and hydrocarbons	Elevated temperature, May require O_2_	Some materials affected by relative humidity
Conductivity: Conducting polymer	No report	High, 0.02 ppm [200]	H_2_, liquid petroleum gases, and CH_4_ for PANI. Other polymers unclear	Room, temperature, May require O_2_	
Work function type: Field effect transistor	No report	Moderate, 54 ppm [206]	H_2_, CH_4_, NO, and ethanol	Elevated temperature, May require O_2_	
Work function type: Metal-oxide-semiconductor capacitor	No report	Moderate, 100 ppm [203]	H_2_	Elevated temperature, May require O_2_	
Work function: the Schottky diode	No report	Moderate, 25 ppm [216]	NO and H_2_	Elevated temperature, May require O_2_	
Thermoelectric	No report	High, 1 ppm [220]	Depend on catalyst	Elevated temperature for best sensitivity, Require O_2_	Require sophisticated signal processing

## 3. Dissolved H_2_ Sensors

### 3.1. Current Status of the Dissolved H_2_ Measurement

Dissolved H_2_ (DH) measurements receive much more attention from researchers because DH concentration is an important parameter in the power industry [14], environmental science [229], and biochemical process [18]. DH concentration is an effective, early-stage indicator of metals and alloys corrosion developments in high temperature aqueous flows in thermal power plants [14,15]. DH concentration in transformer oil also provides valuable information to diagnose the transformer operating condition [230]. In environmental science, DH concentration (around 1–3 mg/L(ppm)) is an important parameter to study thermodynamic equilibria and kinetics in hydrothermal system [229] and redox reactions of sediments microbial dissolution in anaerobic aqueous environments [16,17]. The precise measurement and control of DH concentration in anaerobic digesters is vital for maintaining microbial activities in biomaterials production process [19,231,232].

Therefore, many dedicate sensors were reported to directly measure DH in aqueous solutions [7,8] and in transformer oils [233,234]. Indirect DH measurement methods, which measure extracted gas phase H_2_ with dissolved gas extraction apparatuses were also reported [18,19]. Membrane coated electrochemical DH sensors are the mostly reported sensors to measure DH in aqueous solutions [8,14,231]. Selectively permeable membranes, such as Teflon^®^ and silicone membrane, were applied to separate sensors’ electrolytes from the liquid samples and to allow the selective diffusion of H_2_ to the electrodes [7]. Two types of H_2_ detection mechanisms were reported for electrochemical DH sensors. The first type of DH sensors can be fabricated using a counter electrode made with silver, working electrode made with platinum, gold or palladium, and aqueous KCl electrolyte [8,235]. The redox reactions between AgCl and H_2_ take place with a suitable polarization voltage (such as 600 mV [236]); therefore, O_2_ is not required in DH measurements:(14)$$2\mathrm{AgCl}+2e^{-}\to 2\mathrm{Ag}+2{\mathrm{Cl}}^{-}$$
(15)$$H_{2}\to 2H^{+}+2e^{-}$$

The second type of DH sensors utilize the oxidation of H_2_ with O_2_ [14]:(16)$$\frac{1}{2}O_{2}+2e^{-}+H_{2}O\to 2{\mathrm{OH}}^{-}$$
(17)$$H_{2}\to 2H^{+}+2e^{-}$$

Since KCl electrolyte is not involved in redox reactions, the DH sensor can be fabricated with solid YSZ (yttria-stabilized zirconia) electrolyte and operated in aqueous solutions up to 300 °C [14]. The counter electrode and the working electrode is made with silver powder and porous platinum/palladium, respectively [14].

Measurements of DH concentration in transformer oil were reported with different work function type sensors [234,236,237] and optical fiber sensors with fiber Bragg gratings (FBGs) [233]. Due to good insulation of transformer oil, the work function type DH sensors measure the voltage shift generated from the dipole layer at the metal (usually palladium)-semiconductor interface [236,237] or the conductivity change from interactions between H_2_ and palladium nanowires [234]. Optical fiber DH sensors exploit the H_2_ absorption property of palladium metal at room temperature to generate a mechanical strain, which can be measured through a wavelength change of transmitted light [233]. However, this detection process is very slow with a response time reported ranged in hours to days [233].

Indirect DH measurement were performed using the similar physical gas extraction methods reported in the dissolved CO measurement, such as the static headspace equilibrium method [17], gas stripping technique [18], and membrane extractor [7]. The extracted, gas phase H_2_ was measured by the common method such as gas chromatography [17], conductivity H_2_ sensor [18], and electrochemical H_2_ sensor [7].

### 3.2. Possible Improved Methods for the DH Measurement

In our research, a commercial electrochemical DH sensor was used to measure DH concentrations in the syngas fermenter. The manufacturer reported that the DH sensor only had selectivity issues from H_2_S gas. However, the complicated operation procedures and limited sensing-tip lifespan (up to six months, according to the manufacturer) impede the sensor’s application in commercial syngas fermentation industries. Thus, it is worthy to find a different H_2_ detection mechanism for further DH sensor fabrication. Hübert et al. [238] provided a comprehensive review of current H_2_ detection methods, which categorized H_2_ detection mechanisms as follows: (1) catalytic; (2) thermal conductivity; (3) electrochemical; (4) resistance; (5) work function type; (6) mechanical; (7) optical; and (8) acoustic sensors.

The unique conditions of syngas fermentation means that only certain H_2_ sensors are suitable. Catalytic and thermal conductivity sensors are inappropriate due to their detection mechanisms based on H_2_ oxidation, which is unacceptable in anaerobic syngas fermentation process. Resistance and work function sensors fabricated with metal-oxide semiconductors also share the same O_2_ issue in anaerobic fermentation [238].

However, the absorption of H_2_ in certain metals, such as palladium (Pd) [239], and alloys (palladium-nickel [240]) can be used for H_2_ detection without O_2_. Hydrogen molecules will occupy the interstitial sites in the metal lattice when absorbed by metals, which results in the expansion of the metal [238]. The expansion further changes the electrical, mechanical, and optical properties of the metal, such as conductivity, volume, and optical transmittance and refractive index, respectively.

Metallic resistor H_2_ sensors utilize the reduction of conductivity by H_2_ absorption on palladium films to detect H_2_. Mechanical sensors were reported using microcantilevers [241] or microswitches [242] with palladium coating to detect H_2_ by the volume change from H_2_ absorption. However, the response time, sensitivity, and selectivity are greatly determined by the fabrication methods and structure of coatings or films. Optical H_2_ sensors, such as micro mirror optical fiber sensors, interferometric sensors, evanescent field interaction sensors, surface plasmon resonance (SPR) sensors, and optical time domain reflectometry sensors, exploit the refractive index change from H_2_ absorbed palladium to detect H_2_ [238].

Fiber Bragg gratings (FBGs) sensors utilize the expansion of palladium by H_2_ related change of the Bragg wavelength to detect H_2_ [243]. The response time of FBG sensors varies from seconds [243] to hours [233] due to the thickness of the palladium sensing layer. FBG optical sensors fabricated with chemochromic materials, such as tungsten oxide (WO_3_) [244,245], are another method to detect H_2_ without O_2_. Tungsten oxide reacts with hydrogen to form tungsten bronze, which results in an increase of absorption in the visible range. Using platinum (Pt) as a catalyst can accelerate the reaction speed dramatically [246]:(18)$${\mathrm{WO}}_{3}+xH_{2}\overset{\mathrm{Pt}}{\to }{\mathrm{WO}}_{3-x}+xH_{2}O$$
(19)$${\mathrm{WO}}_{3-x}+\frac{x}{2}O_{2}\overset{\mathrm{Pt}}{\to }{\mathrm{WO}}_{3}$$

The response of these tungsten oxide sensor are significantly faster than FBG sensors fabricated with palladium, which only requires a few seconds at room temperature [245]. The disadvantage is that the recovery of tungsten bronze (WO_3-x_) to tungsten oxide (WO_3_) requires O_2_ as a reagent.

Metallic resistor H_2_ sensors and FBG H_2_ sensors fabricated with palladium were reported in for direct DH measurements in transformer oil [233,247]. However, palladium coatings are subjected to mechanical damage, such as cracking, blistering, and delamination, when repeated exposure to H_2_ [245]. The long-term stability of palladium is also questionable as the metal can be poisoned by CO and CH_4_ [238]. Their inferior sensitivity in a low concentration and humid environment implies that palladium sensors may not work as expected for syngas fermentation, where the DH concentration is very small (less than 3 ppm) and the relative humidity close to 100%. In summary, these O_2_-free H_2_ sensors require more development to replace the current electrochemical DH sensors.

## 4. Discussion and Summary

Real-time, precise measurement and control of dissolved CO (DCO) and H_2_ (DH) concentrations in the fermentation medium is considered as one of the important prerequisites for stable operation of large-scale syngas fermentation fermenters [1,30]. While DH concentration can be readily measured with commercial electrochemical sensors, automatic inline DCO measurement is still an unsolved problem in syngas fermentation.

Review of current CO detection mechanisms implies that fluorescent sensors fabricated with the palladium (Pd) catalyzed Tsuji–Trost reaction [74] or transition metal complexes [68] might be the most feasible method to directly detect DCO concentrations in the syngas fermentation medium. These sensors demonstrate a low detection limit, high CO sensitivity, and good selectivity to gases and chemicals [73,81]. These fluorescent sensors have a promising prospect, as they have been reported for CO measurements in living tissues and aqueous solutions [32,68]. However, repeated DCO measurement with fluorescent sensors remain challenging at the current condition because of problems in sensor reversibility and response time, fermentation interference, and ultraviolet inhibition on microorganisms.

A more practical method is to measure DCO concentration by two sequential steps: (1) extraction of dissolved phase CO from liquid samples and (2) measurement of gas phase CO to calculate DCO concentration. This indirect method can be built with various physical gas extraction methods, such as the gas stripping technique [39], static headspace equilibration method [40], and vacuum extraction systems [41]. For the gas phase CO measurement, infrared and photoacoustic sensors are considered the best methods for indirect DCO measurement in the syngas fermentation environment. However, because of the complicity of the dissolved gas extraction process, the indirect DCO measurement methods suffer a slow response and complicate system setup.

Currently, commercial membrane coated DH sensors based on the electrochemical [7,8] detection mechanism are available on the market. Due to their complicated, laborious operation procedures and limited lifespan, these electrochemical sensors are still not ideal for large-scale application in syngas fermentation. Review of the H_2_ detection mechanisms suggests that palladium based H_2_ sensors might be capable to replace electrochemical DH sensors. Utilizing the absorption of H_2_ in palladium, it provides an approach to measure H_2_ in room temperature without O_2_. Metallic resistor H_2_ sensors and FBG H_2_ sensors fabricated with palladium were reported in for direct DH measurements in transformer oil [233,247], which suggests their potential application in aqueous solutions. Nonetheless, the practical application of DH sensors based on palladium requires further improvement on their sensitivity, long-term stability, response time, and mechanical strength.

In summary, both direct and indirect measurement methods are feasible for DCO measurement in syngas fermentation, using fluorescent sensors and infrared/photoacoustic sensors with dissolved gas extractor, respectively. However, the indirect method has the least technology obstacles currently albeit with a slow response and complicated sensor structure. For DH measurements, H_2_ sensors fabricated with palladium might be able to replace the current electrochemical DH sensors, although many practical challenges from the sensing materials need to be resolved before their widespread applications.

## Figures and Tables

**Figure 1 sensors-21-02165-f001:**
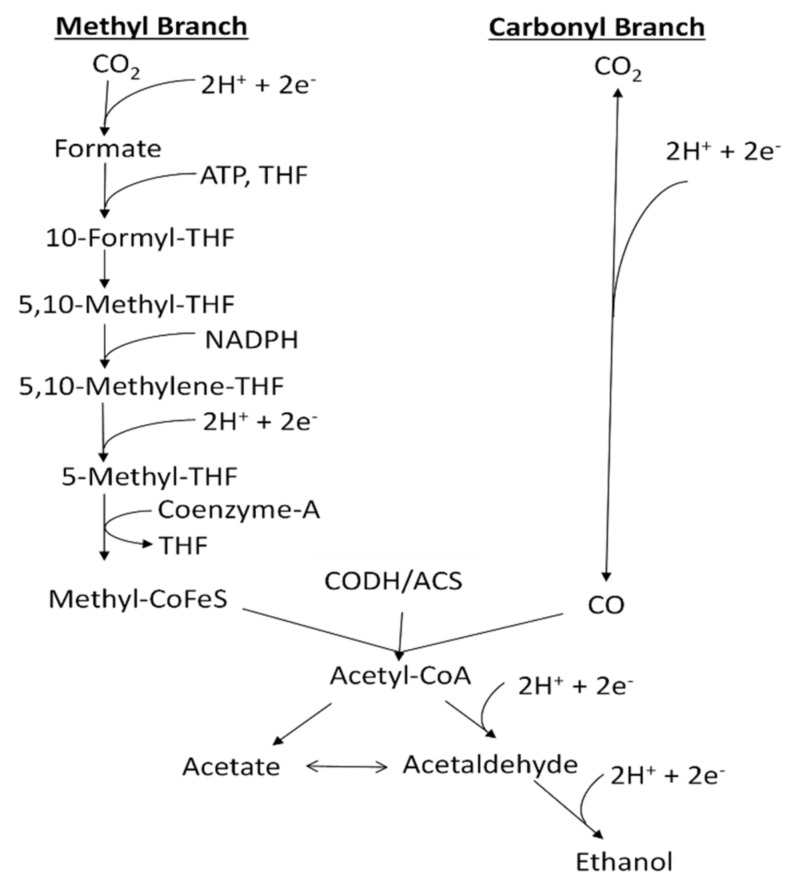
Acetyl-CoA pathway (Wood–Ljungdahl pathway) for the production of ethanol from CO, adapted from [1].

**Figure 2 sensors-21-02165-f002:**
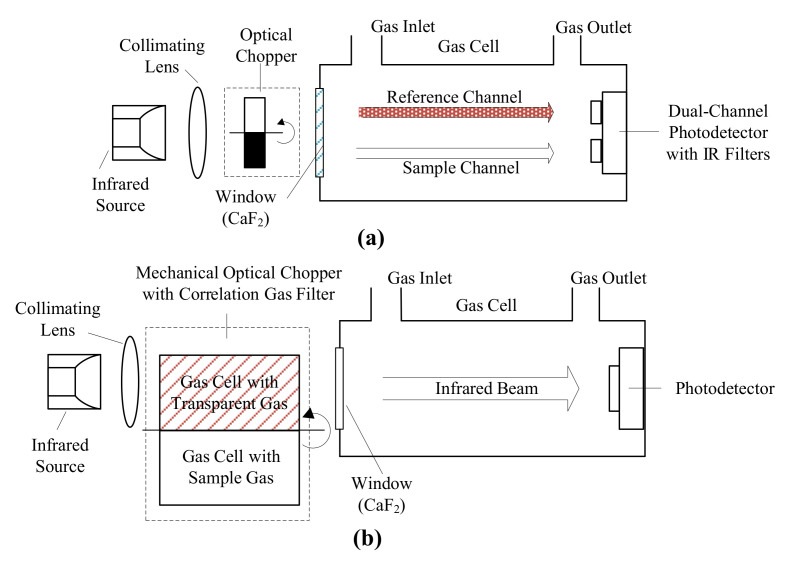
Scheme of nondispersive infrared (NDIR) sensors: (**a**) double beam NDIR sensor structure, adapted from [88] and (**b**) single beam NDIR sensor structure with a correlation gas filter technique, adapted from [99].

**Figure 3 sensors-21-02165-f003:**
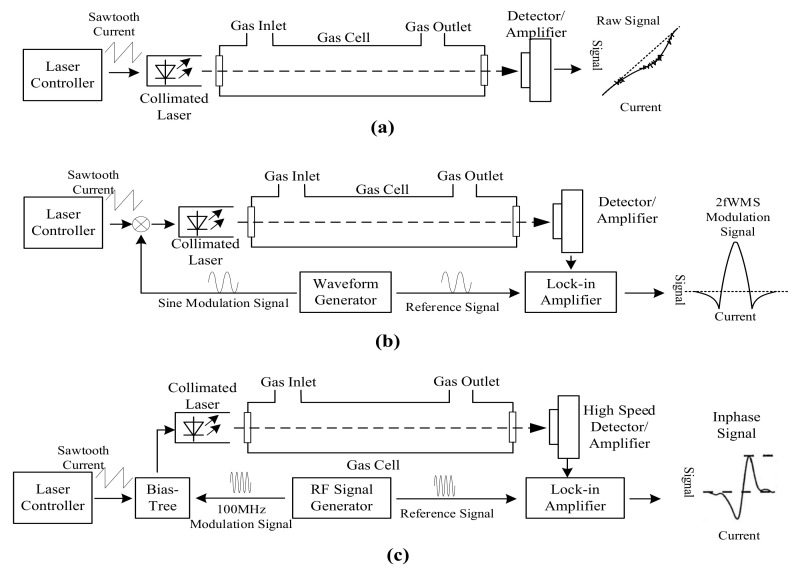
Scheme of tunable diode laser absorption spectroscopy (TDLAS) sensors, adapted from [101,132]: (**a**) TDLAS based on direct absorption spectroscopy; (**b**) wavelength modulation spectroscopy (WMS) TDLAS; and (**c**) frequency modulation spectroscopy (FMS) TDLAS.

**Figure 4 sensors-21-02165-f004:**
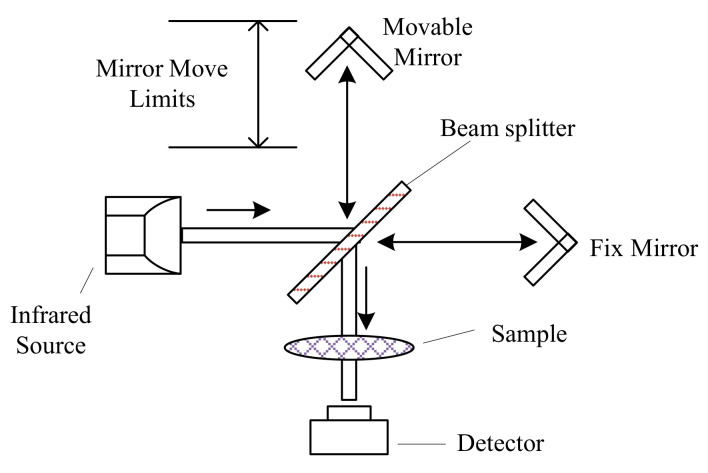
Scheme of Fourier transform infrared (FTIR) spectrometers using the Michelson interferometer, adapted from [88].

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
