# Peer review of "Review of Dissolved CO and H2 Measurement Methods for Syngas Fermentation"

_sensors, 2021, doi:10.3390/s21062165_

Round 1
Reviewer 1 Report
The authors present a review of different sensor types that are available for the
Relevant concentration range should be given in the introduction and should be given as is standard in gas sensor research in ppb/ppm. The authors only state the concentration in line 583 currently which is far to late in the manuscript. Useful to have a scheme to help the largely sensor specialized audience of the journal to understand the syngas fermentation process. Here information about how the sensors could be added to the system can be given. Also here the importance of stability should be reiterated ( how often could such a sensor system be replaced). Price is also an important consideration. How expensive could a sensor setup be to still be considered feasible for this application?
Some of the sentences are clumsy making it difficult for the reader to follow. The authors should take time to reread the manuscript and simplify sentences when possible. For example, line 145-146 ‘’Electrochemical sensors measure changes in electrical properties from CO’s electrochemical reactions occurred at the sensors’ electrodes to detect CO.’’ is confusing and doesn’t reveal much about the operation principle. A more detailed description should be added. Similar to the description of Hunter et al. https://iopscience.iop.org/article/10.1149/1945-7111/ab729c.
The authors over-simplify the origin of the linearity between the sensor response of amperometric gas sensors 151-153. More detail should be added as to why the response is linear, similar to the review by Helm et al. https://www.ncbi.nlm.nih.gov/pmc/articles/PMC3292126/.
The authors state in line 166 that, ‘’The electrodes materials of amperometric CO sensors determine sensor CO selectivity.’’ This is true but the cell can additionally be optimized based on the potential of the working electrode and through the addition of chemically active filters (https://onlinelibrary.wiley.com/doi/pdf/10.1002/elan.1140040302).
The description of the disadvantages for the amperometric sensors is also too brief and simplistic. The authors should describe why operation in dry air is problematic for electrochemical sensors. The authors argue that due to the anaerobic conditions, amperometric gas sensors may not be the most appropriate sensors. Are they at all applicable in anaerobic conditions? If not then the authors should streamline this section to provide the reader a clear understanding why amperometric sensors are not suited for this application. Overall the section very general and should be tuned for the topic discussed in the review, dissolved CO and H2 measurement methods for syngas fermentation. The same applies to potentiometric gas sensors.
In the section on metal oxide gas sensors, the authors cite Rb2O as a typical sensor material for gas sensors ( the citation is on In2O3 with Rb2O as an additive). This is a crucial difference! Rb2O serves as activator for the interaction between In2O3 and CO and is not the sensitive material! This authors should carefully review this section. The authors also do not mention WO3 which is after SnO2, of the most common base materials for metal oxide based gas sensors.
The explanation for the operation principle of the metal oxide sensors is incredibly brief and not very informative, 202, ‘’The conductivity changes of most metal oxides can be explained as the result of the electrons trapping and the band bending of the oxides associated with reactions between CO and oxygen.’’ Here it must be explained if it is an n-or p-type material, what happens and how this changes with CO.
Again also for metal oxide gas sensors the reaction with atmospheric oxygen is important. Although more robust than electrochemical sensors (can be operated in dry air), the question remains is in anaerobic conditions these sensors are at all suited? Also in the introduction the authors mention the formation of ethanol. Most n-type metal oxide based gas sensors are known to significantly respond to ethanol vapor!
The section on conducting polymer semiconductor sensors is very brief. Interestingly in their review, Bai and Shi report that the stable resonance structure of +C≡O− with the positive charge at the carbon atom will withdraw a lone pair electrons at the amine nitrogen: −NH−. https://www.mdpi.com/1424-8220/7/3/267/htm This would mean that theoretically conducting polymer sensors could be suited to for anaerobic conditions? The authors should expand on this section and discuss the potential of the sensor for the application. Also, although limited research might exist, the authors should provide information about the stability of the sensors in application relevant conditions.
The same as for the previous sensors is again is true for the FETs., diodes, MOS capacitors, acoustic waves. The operation principles are not adequately described and issues of stability/applicability are not covered.
‘’Thermoelectric sensors utilize the catalytic oxidation of CO to measure CO concentration.’’ Also these sensors are poorly suited for the application.
The operation principle of Vacuum ultraviolet resonance fluorescence sensors is especially vague. Based on the presented information, however, these sensors may be suited for the application. How expensive are these sensors? Can they discriminate the CO concentrations in the desirable range (sensitivity fitting)? How long can they be reliably operated/do they need to be recalibrated?
‘’ Transition metal oxides sensors have inferior selectivity to H2, because the selected metals also 341 trigger the catalytic oxidation or absorption of H2 [117,120,121]. The selectivity of these sensors to 342 other gases are also unclear.‘’ Are the authors’ sure that this statement is accurate. Noble loaded WO3 is known to be an excellent material for the optical detection of H2! This section must be reconsidered and a more thorough review of literature should be done.
The section on infrared sensors is more thorough than the other detection methods. Indeed the authors make the argument that this method is best suited for the application. In the introduction the authors argue that a detection method is needed that would allow for in line measurements to be done. How would this be realized using an infrared based sensor? How long would an accurate measurement take? Also here is the concentration level high/low enough to make the method applicable? How expensive is an appropriate setup? How about the long term stability?
The chart that shows that shows the sensors and their information in different categories, is very good. It should include both CO and H2 sensors, as that is the focus of the review paper. It should be extended to include price, long term stability and ease of operation. Maybe the authors could color coordinate the sensors based on their applicability.
Overall the review is incredibly descriptive and not critical. In its current state it provides mostly shallow information. The description of H2 and CO Due to the very specific topic of the review article, the authors must more critically discuss the problematic aspects of the measurements. Ideally the review should focus more on the detection methods that are promising for the application. In fact, instead of briefly mentioning all the different methods, the authors could instead refer to various sources on each method that is poorly suited (even just a citation in the sensor overview chart) and then concentrate more thoroughly on the better suited detection methods. The weaknesses/ strengths of each method could then be described in more detail. Additionally more information about how inline measurements should be done could must be provided.
Author Response
Dear Reviewer 1,
On behalf of all the authors, I would like to sincerely appreciate your valuable comments on the manuscript. Your comments provide valuable opinion to improve the quality of the manuscript. We also benefit from them for our future research.
Based on your review comments, we made a major revision not only in the contents, but also in the structure of the original manuscript. We wish with our efforts, the manuscript was improved at a better quality.
In the following pages, we described the changes made corresponding to each comment. The page number and the line number indicated in this document referred to the revised manuscript.
Please let us know if there are any further questions.
Best Regards,
Sincerely,
Jie Dang

Reviewer 2 Report
Many types of gas sensors are used to detect the gas under the air. If the concentration of oxygen in the air changes, the signal of the sensor will change greatly. These sensors include: electrochemistry, metal oxide resistance, field effect, catalytic combustion, vibration mass. The optical type is more likely to be used for measuring dissolved CO and H2 during syngas fermentation.Author Response
Dear Reviewer 2,
On behalf of all the authors, I would like to sincerely appreciate your valuable comments on the manuscript. Your comments provide valuable opinion to improve the quality of the manuscript. We also benefit from them for our future research.
Based on your review comments, we made a major revision not only in the contents, but also in the structure of the original manuscript. We wish with our efforts, the manuscript was improved at a better quality.
Please let us know if there are any further questions.
Best Regards,
Sincerely,
Jie Dang

Reviewer 3 Report
I think this review is very good. I have ever believed that I was reading a published paper. In line 524 and 525, it is said that the advantage of photoacoustic sensor over infrared sensors is that they did not require highly sensitive photodetector to measure infrared absorption. To my best knowledge, 1ppm of detection limit can be obtained with common Fourier Transform Infrared Spectroscopy. At the same time, photoacoustic sensor need highly sensitive acoustic detector to measure acoustic wave.
Author Response
Dear Reviewer 3,
On behalf of all the authors, I would like to sincerely appreciate your valuable comments on the manuscript. Your comments provide valuable opinion to improve the quality of the manuscript. We also benefit from them for our future research.
Based on your review comments, we made a major revision not only in the contents, but also in the structure of the original manuscript. We wish with our efforts, the manuscript was improved at a better quality.
Please let us know if there are any further questions.
Best Regards,
Sincerely,
Jie Dang

Round 2
Reviewer 1 Report
The manuscript has significantly improved through the revisions. In its present form the article is publishable but would benefit from an English proofreading.
I also have a few small comments:
The authors still dedicate a disproportionately large section of the manuscript to IR sensors. It should more clearly be stated that this method is currently the most promising and therefor will be described in greater detail. Additionally it would be more logical to sort the less suited optical methods together (either all before or after IR sensors).
Line 613 is still misleading as the cited sensor is not based on Rb2O3 but on In2O3 with a Rb2O3 catalyst.
The conclusion, ‘‘A more practical method in current condition is to measure DCO concentration by two sequential steps: extraction of dissolved phase CO from liquid samples and measurement of gas phase CO to calculate DCO concentration. This indirect method can be built with various physical gas extraction methods, such as gas stripping technique [39], static headspace equilibration method [40], and vacuum extraction systems [41]. It also virtually allow any CO sensor to be used for CO measurements.‘‘ is a bit confusing. Particularly the virtually any sensor, is unclear. One of the major limitations hindering the use of several sensor types are the anaerobic fermentation conditions. Is this not a limitation in the case of the indirect extraction method? This point must be clarified.
At 24 pages the review paper is now 4 pages over what is typically set as a maximum by MDPI sensors. Here it would be possible to reduce the length of the paper by streamlining the description of the ill suited detection methods.
Author Response
Response to Review #1
Dear Reviewer 1,
On behalf of all the authors, I would like to sincerely appreciate your valuable comments on the manuscript. Your comments provide valuable opinion to improve the quality of the manuscript. We also benefit from them for our future research.
We described the changes made corresponding to each comment below. The page number and the line number indicated in this document referred to the revised manuscript, ReviewOfCOH2SensorForSyngasFermentation_R3.docx.
Please let us know if there are any further questions.
Best Regards,
Sincerely,
Jie Dang
---------------------------------------------------------------------------------------------------------------------
Reviewer #1:
The manuscript has significantly improved through the revisions. In its present form the article is publishable but would benefit from an English proofreading.
Authors’ Response: Thank for your comments. We carefully went through the manuscript and cleaned the long and unclear sentences.
I also have a few small comments:
The authors still dedicate a disproportionately large section of the manuscript to IR sensors. It should more clearly be stated that this method is currently the most promising and therefor will be described in greater detail. Additionally it would be more logical to sort the less suited optical methods together (either all before or after IR sensors).
Authors’ Response: Based on your comments, we add a paragraph to state the IR sensor as the most promising method. Line 343-346.
Line 613 is still misleading as the cited sensor is not based on Rb2O3 but on In2O3 with a Rb2O3 catalyst.
Authors’ Response: Corrected the name of catalysts. Line 597-602.
The conclusion, ‘‘A more practical method in current condition is to measure DCO concentration by two sequential steps: extraction of dissolved phase CO from liquid samples and measurement of gas phase CO to calculate DCO concentration. This indirect method can be built with various physical gas extraction methods, such as gas stripping technique [39], static headspace equilibration method [40], and vacuum extraction systems [41]. It also virtually allow any CO sensor to be used for CO measurements.‘‘ is a bit confusing. Particularly the virtually any sensor, is unclear. One of the major limitations hindering the use of several sensor types are the anaerobic fermentation conditions. Is this not a limitation in the case of the indirect extraction method? This point must be clarified.
Authors’ Response: Revised. Line 835-842.
At 24 pages the review paper is now 4 pages over what is typically set as a maximum by MDPI sensors. Here it would be possible to reduce the length of the paper by streamlining the description of the ill suited detection methods.
Authors’ Response: Thanks for the suggestion. We revised the figures (Figure.2-4) to cut some length, while revised some content to save space.

Reviewer 2 Report
After a round of modification, this paper has made a detailed analysis of the requirements for the use of gas detection technology in syngas promotion, especially the interference of oxygen on CO and H2. However, the structure and layout of the paper are quite chaotic, with many subtitles. It is suggested that the paper be published after revision
Author Response
Response to Review #2
Dear Reviewer 2,
On behalf of all the authors, I would like to sincerely appreciate your valuable comments on the revised manuscript. Your comments provide valuable opinion to improve the quality of the manuscript.
Based on your review comments, we revised the structure of the revised manuscript. We wish with our efforts, the manuscript was improved.
Please let us know if there are any further questions.
Best Regards,
Sincerely,
Jie Dang
---------------------------------------------------------------------------------------------------------------------
Reviewer #2:
After a round of modification, this paper has made a detailed analysis of the requirements for the use of gas detection technology in syngas promotion, especially the interference of oxygen on CO and H2. However, the structure and layout of the paper are quite chaotic, with many subtitles. It is suggested that the paper be published after revision
Authors’ Response: Thanks again for reviewing our revised manuscript. During the second revision, we revised the paper headings and reduced the subtitles to make the paper better organized.